# Molecular Cloning, Characterization, and Expression of a Receptor for Activated Protein Kinase C1 (RACK1) Gene in *Exopalaemon carinicauda* Zoea Larvae under Aroclor 1254 Stress

**DOI:** 10.3390/biology13030174

**Published:** 2024-03-08

**Authors:** Yuefeng Cai, Jie Hu, Yepeng Guo, Xin Shen

**Affiliations:** 1Jiangsu Key Laboratory of Marine Bioresources and Environment/Jiangsu Key Laboratory of Marine Biotechnology, Jiangsu Ocean University, Lianyungang 222005, China; 13815743367@163.com (J.H.); guoyepeng6@163.com (Y.G.); shenthin@163.com (X.S.); 2Co-Innovation Center of Jiangsu Marine Bio-Industry Technology, Jiangsu Ocean University, Lianyungang 222005, China

**Keywords:** receptor for activated protein kinase C1, *Exopalaemon carinicauda*, clone, characteristic, environmental stress, expression response

## Abstract

**Simple Summary:**

The *Ec*RACK1 protein from *Exopalaemon carinicauda* has been identified and analyzed. It consists of 957 nucleotides, encoding 318 amino acids. *Ec*RACK1 exhibits a molecular weight of approximately 35.6 kDa and a theoretical isoelectric point (pI) of 7.90. Moreover, the secondary structure analysis reveals the presence of seven WD repeats in *Ec*RACK1, which exhibit more than a 96% sequence identity with the RACK1 protein found in *Penaeus species*. Additionally, significant upregulation of *Ec*RACK1 expression is observed during stages II and IV, with tissue-specific expression detected in hepatopancreas, spermary, and muscle tissues of adult *E. carinicauda* individuals. Furthermore, exposure to Aroclor 1254 for durations of 6, 10, 20, and 30 days significantly induces the expression level of *Ec*RACK1 compared to the control group (*p* < 0.05). These findings suggest that *Ec*RACK1 may play crucial roles in larval development as well as environmental defense mechanisms in *E. carinicauda*.

**Abstract:**

The receptor for activated protein kinase C1 (RACK1) belongs to the typical WD repeat family, which is extremely conservative and important in multiple signal transduction pathways related to growth and development that coordinate the intracellular role of various life activities. As a novel protein with versatile functions, it was found in a variety of organisms. In a previous study, we identified the RACK1 sequence of white shrimp from transcriptome data. In this study, we employed specialized bioinformatics software to conduct an in-depth analysis of *Ec*RACK1 and compare its amino acid sequence homology with other crustaceans. Furthermore, we investigated the expression patterns of RACK1 at different developmental stages and tissues, as well as at various time points after exposure to Aroclor 1245, aiming to elucidate its function and potential response towards Aroclor 1245 exposure. The length of *Ec*RACK1 is 957 nucleotides, which encodes 318 amino acids. Moreover, there were seven typical WD repeats in *Ec*RACK1, which have more than a 96% sequence identity with the RACK1 proteins of *Penaeus*. The results of tissue expression and spatiotemporal expression showed that it was significantly increased in the II and IV stages, but had a significant tissue specificity in the hepatopancreas, spermary, and muscle tissues of *E. carinicauda*, adult stage. Compared to the control, *Ec*RACK1 was significantly induced in *E. carinicauda* zoea larvae exposed to Aroclor 1254 for 6, 10, 20, and 30 d (*p* < 0.05). These results suggested that *Ec*RACK1 may play an important role in the larval development and environmental defense of *E. carinicauda*.

## 1. Introduction

The widespread presence of a scaffolding protein receptor for activated protein kinase C1 (RACK1), which exhibits regulatory activity with protein kinase C (PKC) in both prokaryotes and eukaryotes, has garnered increasing attention and scrutiny [1,2]. As a highly conserved member of the tryptophan-aspartic acid domain (WD) repeat protein family, RACK1 participates in many biological processes, such as proliferation, cell migration, apoptosis, development, cellular adaptation processes, and immune responsiveness [1,3,4,5,6,7,8]. In addition, RACK1 is known as an adaptor protein in multiple intracellular signal transduction pathways, such as the PKC/RACK pathway [9], the cyclic AMP-specific phosphodiesterase pathway [10,11], the tyrosine kinase-phosphatase pathway [12], and the Aryl hydrocarbon receptor (AhR) signaling pathway [13,14].

Numerous homologues of RACK1 have been preliminarily investigated in various aquatic invertebrates and were demonstrated to possess diverse functionalities. The conserved seven WD repeats characteristic of the RACK1 protein have been identified across several invertebrate species, including silkworms, crabs, shrimps, and mollusks. Notably, studies on *Chlamys farreri* [15], *Mya arenaria* [16], *Mytilus coruscus* [17], *Bombyx mori* [18], *Eriocheir sinensis* crabs [19], *Fenneropenaeus chinensis* shrimp [20], *Penaeus monodon* black tiger shrimp [21], and *Pacifastacus leniusculus* freshwater crayfishes [22] reveal a conserved function and role for the RACK1 gene family.

The involvement of RACK1 in immune defense and abiotic stress responses has been demonstrated in shrimp and mussel research [20,21,23,24]. In hemocytes of *Penaeus vannamei* and *P. monodon*, the expression level of the RACK1 gene significantly increased after injection with *Vibrio alginolyticus*, suggesting a significant impact on the innate immunity of shrimp. Additionally, RACK1 has been implicated in tissue differentiation and cell growth during the larval development of oyster *Crassostrea angulata* [25]. Furthermore, in the freshwater crayfish *P. leniusculus*, RACK1 functions as a negative feedback regulator of the circadian clock [22].

According to previous studies, RACK1 has been implicated in the response to oxidative stress induced by bacteria and cadmium in pearl oyster *Pinctada martensii* [26], tributyltin (TBT) in *M. arenaria* [16], and Benzo[a]pyrene (BaP) in *C. farreri* [13,15]. In *M. coruscus*, the regulation of silently expressed RACK1 led to increased activities of superoxide dismutase (SOD) and catalase (CAT) in hemocytes, suggesting a crucial role for RACK1 in the oxidative stress response of *M. coruscus* [17].

In our previous study, RACK1 was demonstrated to play a crucial role in the metabolic pathways of exogenous substances in the invertebrate scallop *C. farreri*. Furthermore, after exposure to BaP, it exhibits the ability to mitigate oxidative stress through its interaction with AhR [13]. Additionally, it was found that the transcriptional expression level of the RACK1 gene in clam *M. arenaria* decreased after exposure to TBT. However, in invertebrates, the detailed mechanism of RACK1 involved in defense and response is still very limited.

Polychlorinated biphenyls (PCBs) exhibit resistance to degradation, possess lipophilic and hydrophobic properties, and demonstrate persistence, accumulation, and long-range migration. Among these compounds, the predominant mixture is Aroclor 1254, which indicates a chlorine content of approximately 54% by mass. These persistent organic pollutants have been detected in oceans and may be utilized by marine organisms [27]. The concentrations of PCBs in Dalian Hutan and Lianyungang were measured at 33.55 ng/L and 65.43 ng/L, respectively [28]. The pollution levels in the Xiamen Dadeng Sea area, Fuzhou Pingtan Sea area, and Zhuhai Qiao Island were 74.51 ng/L, 416.80 ng/L, and 474.92 ng/L, respectively [29]. Therefore, PCBs pose a potential threat to marine organisms.

The susceptibility of aquatic organisms to PCBs may be heightened due to their increased potential for bioaccumulation and biomagnification [30]. In particular, aquatic animals can experience a multitude of adverse effects including growth inhibition, immunosuppression, hepatotoxicity, reproductive and developmental toxicity, endocrine disruption, as well as the induction of cytochrome P450 enzymes [31]. Marine organisms in the juvenile stage are highly vulnerable to foreign pollutants due to their incomplete tissue development. Numerous studies have demonstrated the significant impact of PCBs on the growth, survival, and developmental processes of juvenile marine organisms. Exposure to PCB concentrations of 500 ng/L and 1000 ng/L resulted in notable alterations in *Neomysis awatschensis* larvae’s total body length, antennal scale length, exopod length, endopod length, and telson morphology [32]. The study investigating the impact of PCBs on the development of *Euphausia superba* larvae revealed a pronounced sexual dimorphism in response to PCB exposure. Therefore, it is important to study the effects of PCBs on the juvenile development of marine organisms [33].

*Exopalaemon carinicauda* is widely distributed in coastal areas and belongs to the benthic shrimp species. Due to its rapid growth, strong reproductive capacity, euryhaline nature, and ease of feeding, it has become a prominent shrimp species bred in recent years. This study presents the gene characteristics, adult tissue distribution, and developmental expression pattern of *Ec*RACK1 in *E. carinicauda* zoea larvae. Furthermore, to explore its potential function in the metabolic pathways of exogenous substances in invertebrates, we measured the mRNA levels of *Ec*RACK1 after exposure to Aroclor 1254. Our research enhances the understanding of the characterization and multiple functions of RACK1 in invertebrates.

## 2. Materials and Methods

### 2.1. Shrimps’ Maintenance and Exposure

The adult *E. carinicauda* shrimps were collected from the Haitou dock in Lianyungang, Jiangsu Province, China. The shrimp length measured 7.0 ± 0.5 cm. Following a stocking density of 5 tails per liter, they were cultured in tanks measuring 60 cm × 45 cm × 35 cm and filled with 30 L of filtered artificial seawater (salinity: 31 psu; pH: 7.8) at a temperature of 18 ± 2 °C. During the experimental period, after being fed the fresh clams *Meremetrix meremetrix*, it was kept at 100% aeration and had a 50% water change every other day.

After one week of acclimation, fresh tissue samples (gill, hepatopancreas, muscle, spermary, and ovary) weighing 80~100 mg each were immediately collected from thirty adults and added to 200 μL of RNAiso Plus (TaKaRa, Dalian, China). Subsequently flash-frozen in liquid nitrogen, the samples were homogenized at a speed of 12,000 rpm/min for 15~30 s. Then, an additional 800 μL of RNAiso Plus was added to ensure complete cell and tissue lysis. Total RNA extraction was performed promptly followed by cDNA synthesis using PrimeScript II RTase reverse transcriptase (PrimeScript™ II 1st Strand cDNA Synthesis Kit, Takara, Dalian, China) at 42 °C for 60 min with adaptor primer oligo (dT), following the manufacturer’s protocol.

The Aroclor 1254 (CAS:11097-69-1; AccuStandard Inc., New Haven, CT, USA) was dissolved in dimethyl sulfoxide (DMSO; Sigma-Aldrich Co., St. Louis, MO, USA) to prepare a stock solution with a concentration of 10 g/L. Subsequently, the Aroclor 1254 stock solution was diluted with DMSO and added to seawater to achieve a final DMSO concentration of 0.001% (*v*/*v*), ensuring no damaging effects on the organism were observed [34]. Female shrimp carrying eggs were carefully selected and hatched under optimal conditions. The newly hatched zoea larvae were collected simultaneously for the exposure experiment. Random allocation divided the larvae into three experimental groups with three replicates each. Two concentrations of Aroclor 1254, namely 10 ng/L and 1000 ng/L, were chosen for exposure. During the treatment period, shrimp larvae were fed fresh nauplius of *Artemia salina* as their diet while maintaining all other conditions identical to those during acclimatization. At time points of 0, 6, 10, 20, and 30 d exposure, thirty larvae from each replicate group were sampled for subsequent analysis immediately after collection.

### 2.2. Cloning the ORF Fragment of EcRACK1

The open reading frame (ORF) fragment of *Ec*RACK1 was amplified by using specific primers designed by transcripts sequence in our transcriptome library (unpublished) (Table 1). The polymerase chain reaction (PCR) was performed in a total volume of 25 μL using Gradient Mastercycler (Eppendorf, Germany), PCR mixture containing reaction buffer with 8.5 μL ddH_2_O, 1 μL of each primer (10 μM), 2 μL template cDNA, and 12.5 μL of PCR Master Mix (Takara, Dalian, China). For *Ec*RACK1 amplification, 3 step-PCR cycles were followed by 35 cycles of 98 °C for 10 s, 54 °C for 30 s, and 72 °C for 1 min. PCR products were analyzed by electrophoresis on 1.0% agarose gel stained with GelRed (Biotium, Fremont, CA, USA) in 1× TAE buffer. The PCR product was purified by using TIANgel Midi Purification Kit (TIANGEN, Beijing, China), and then the two strands were sequenced.

### 2.3. Sequence Analysis of EcRACK1

Sequence analysis and database alignment were performed by using DNASTAR5.0 software (DNASTAR, Inc., Madison, USA) and BLASTx search of the GenBank database (https://www.ncbi.nlm.nih.gov/genbank/, accessed on 25 August 2023). The characteristics of molecular weight and theoretical pI of the *Ec*RACK1 protein were predicted by using SWISS-MODEL (http://swissmodel.expasy.org/, accessed on 25 August 2023). The secondary structure predicted and 3D protein structure models of WD40 domains of the *Ec*RACK1 amino acid sequence were generated by Jpred Server 4.0.0 (https://www.compbio.dundee.ac.ku/JalviewWS/services/jpred/, accessed on 25 August 2023) and SWISS-MODEL Workspace (http://swissmodel.expasy.org/Workspace/, accessed on 25 August 2023). Multiple protein sequence alignment was performed by Clustal Omega (https://www.ebi.ac.uk/Tools/msa/clustalo/, accessed on 25 August 2023), and graphical editing was performed using the Jalview program. The unrooted trees were built by the Molecular Evolutionary Genetic Analysis 5 program (http://www.megasoftware.net, accessed on 25 August 2023) using the Neighbour-joining method, consisting of 10,000 trials with bootstrap.

### 2.4. Expression Analysis by qRT-PCR

The primers for quantitative real-time PCR (qRT-PCR) were obtained using the Primer5.0 software (Primer software, San Francisco, CA, USA) according to the ORF fragment of *Ec*RACK1 (Table 1). The template cDNA were from five tissue samples, six periods of zoea larvae, and samples of the exposure experiment, respectively. The β-actin gene was selected for internal reference. The qRT-PCR was performed using SYBR Premix Ex Taq II (Tli RNaseH Plus) (TaKaRa, Dalian, China) on the ABI StepOne Plus real-time PCR system (PerkinElmer Applied Biosystems, Foster city, CA, USA). The total volume was 20 µL containing 10 µL of 2 × SYBR Premix Ex Taq II (Tli RNaseH Plus), 1 µL of cDNA, 0.8 µL of forward and reverse primers, 7 µL of dH_2_O, and 0.4 µL of 50 × ROX reference dye (TaKaRa, Dalian, China). The reactions were performed using the following conditions: initial denaturation at 95 °C for 30 s, followed by 40 cycles of 95 °C for 5 s, 55 °C for 30 s, and 72 °C for 30 s. Each gene was amplified using three replicates. The relative quantification of each gene was performed according to the mean normalized expression (MNE) method [27].

The data analysis was performed using SPSS Statistics 25 (SPSS Inc., Chicago IL, USA). RACK1 expression was quantified using the 2^−ΔΔCt^ method. All results are presented as means ± SD. One-way analysis of variance with the LSD method was employed for intergroup comparisons. Statistical significance was defined as *p* < 0.05.

## 3. Results

### 3.1. Sequence and Characterization of EcRACK1

In the present study, the complete ORF fragment of the RACK1 gene from *E. carinicauda* was obtained, which contained 957 bp nucleotide bases and encoded 318 amino acids (Figure 1). The sequence was submitted to the National Center for Biotechnology Information (NCBI) database and the GenBank accession number was MH218847.

In addition, the molecular weight and theoretical pI of the predicted protein according to the RACK1 gene from *E. carinicauda* were 35.6 kDa and 8.03, respectively. Moreover, there are seven WD40 repeat domains distributed in the secondary structure of the *Ec*RACK1 protein (Figure 2 and Table 2), including seven blades of a propeller structure and four anti-parallel β-sheets (Figure 3).

As the result of multiple alignments, RACK1 proteins from various species were highly conservative and had more than a 90% similarity with other shrimps and crabs, such as *P. vannamei* (96.86%, XP_027232761.1), *P. leniusculus* (96.86%, AFV15799.1), *E. sinensis* (96.54%, ALF44683.1), and *Scylla paramamosain* (94.65%, AID16303.1), and the similarity between *Ec*RACK1 and other homologues from distantly related species was more than 70% (Appendix A).

Phylogenetic analyses showed that *Ec*RACK1 with other crustaceans’ and arthropods’ RACK1 were clustered together and grouped as a clade with other species of RACK1. The RACK1 of vertebrates, molluscs, and echinoderms were grouped as an independent branch, respectively (Figure 4).

### 3.2. Specific Expression Patterns of EcRACK1

The tissue-specific expression of the *Ec*RACK1 homologue in five different shrimp tissues was assessed by qRT-PCR (Figure 5), and the *Ec*RACK1 homologue transcripts were expressed in all tissues and zoea larvae examined. Compared with the gill and ovary, the expression of *Ec*RACK1 homologues in spermary, hepatopancreas, and muscle tissue is higher, and the lowest mRNA levels are observed in the gills.

### 3.3. Expression Analysis of EcRACK1 under Aroclor 1254 Exposure

The mRNA levels of *Ec*RACK1 were assessed using qRT-PCR in the zoea larvae of *E. carinicauda* that were exposed to Aroclor 1254 (Figure 6). Compared to the control group, the expression went straight from 1 to 6 days and then decreased over the following days (*p* < 0.05). Under the exposure of 1000 ng/L Aroclor 1254, the expression of RACK1 initially increased and subsequently decreased, reaching its maximum on day 10. Meanwhile, after exposure to Aroclor 1254 for 6, 10, 20, and 30 d, the expression of *Ec*RACK1 was significantly induced in *E. carinicauda* zoea larvae (*p* < 0.05). Microscopic observations confirmed that all larvae progressed into the mysis stage. Notably, on the 30th day of exposure when all zoea larvae had reached the late larval stage, there was an extremely significant induction in the expression of *Ec*RACK1 mRNA (*p* < 0.05).

## 4. Discussion

The *Ec*RACK1 gene was cloned and deposited in the NCBI database (MH218847) in this study. Additionally, we presented the characterization, tissue distribution, and developmental expression pattern of *Ec*RACK1 in *E. carinicauda* zoea larvae. In organisms, RACK1 functions as a scaffold protein with versatile roles and plays crucial regulatory roles in the growth and development of plants and animals. It consists of a seven-bladed tryptophan-aspartic acid domain or WD40 repeat anchoring protein that exhibits key regulatory functions in the stress response and cell signal reception [35,36]. RACK1 was initially cloned from cDNA libraries of human B-type lymphocytes and raw chickens [37]. To date, RACK1 has been cloned and sequenced in several crustacean species including *Penaeus japonicas* (Genbank ID: AHF21001.1), *P. monodon* (Genbank ID: ABU49887.1), *P. vannamei* (Genbank ID: AHX56189.1), *S.paramamosain* (Genbank ID: AID16303.1), and *P.leniusculus* (Genbank ID: AFV15799.1). A single RACK1 is recognized in animals while three isoforms are found in plants known as RACK1A, RACK1B, and RACK1C, but both have heterotrimeric G proteins, a β subunit, and a seven-blade propeller structure with an anchoring bracket [36]. Sequence homology suggests that RACK1 may have similar functions across different organisms with over 96% similarity at the protein level. It shows the highest homology with arthropod Crustacea Decapoda, and its genetic similarity to other species is also above 70%, demonstrating its high conservation.

In functional research, RACK1, an intracellular receptor protein, was initially identified as a mammalian anchoring protein for kinase C. Furthermore, it was discovered that RACK1 forms a stable complex with active protein kinase C (PKC) in the rat brain, thereby facilitating downstream signaling pathways and earning the designation activated protein kinase receptor 1 gene RACK1. Subsequently, it was observed that RACK1 in rats can also activate various protein kinases such as PKCβII, PKCа, and PKCε [38]. As a prototypical mediator protein, RACK serves as both a conduit for transmitting signals to target cells during PKC activation and a versatile scaffold molecule capable of recruiting specific proteins involved in diverse signaling pathways with distinct biological functions including neuromodulation, signal transduction, immune repair mechanisms, cell growth regulation, cell adhesion processes, and apoptosis.

In the human hippocampal signaling pathway, RACK1 is regulated by androgen and cortisol. Disruption of the RACK1/PKC signaling pathway could lead to impairments in brain memory formation, which hinders human cognitive learning [39]. Within the RACK1-related Hedgehog signaling pathway, the transition from a complex consisting of Cubitus interruptus/RACK1/Costal2 to Smoothened/RACK1/Costal2 results in an increased susceptibility to Hedgehog-related diseases. This regulatory mechanism has been found to be highly conserved across mammalian cells and drosophila [40]. Furthermore, it has been observed that the transcription and expression of the RACK1 gene are influenced by androgen receptors, while estrogen induces RACK1 expression through G-protein-coupled estrogen receptor (GPER) 1.

RACK1, a receptor involved in the signal transduction of steroid active compounds, plays a crucial role in the transmission of sex hormones. It exerts regulatory effects on immune cell function, impacts the initiation of immune responses, maintains peripheral immune tolerance to autoantigens, and offers the potential for immunoactive substances related to sex hormones [41].

The developmental stages of *E. carinicauda*, including egg, flea larvae, and adult, were selected for analysis of fluorescence quantification in six stages of zoea larvae of *E. carinicauda*. It was observed that the RACK gene exhibited expression in all six stages without significant differences, indicating its crucial role in the development of juveniles in *E. carinicauda*. Furthermore, a quantitative analysis of the gene was conducted in various tissues of adult *E. carinicauda* individuals. In eggs, the gene expression level was significantly higher compared to adult tissues, suggesting a potential involvement of RACK1 in larval development. Notably, during the embryonic development period, there was an increase in RACK1 content at the early-stage eggs of *E. carinicauda*, which provides a theoretical foundation for comprehending and investigating the molecular mechanism underlying EcRACK1’s impact on growth dynamics within *E.carinicauda* larvae.

In previous studies, RACK1 proteins have been identified as multifunctional proteins with high biological conservation across different species [42,43,44]. Further investigation revealed the presence of seven repeated WD40 structures in the protein, forming seven beta propeller structures that serve as central channels for protein binding in both prokaryotes and eukaryotes. Different regions of RACK1 are involved in various organism processes, with multiple binding sites present within different WD40 repeat regions. The first to second WD40 repeats contain a binding site for the Na^+^/H^+^ exchanger ISOform 5 (NHE5), which regulates ion balance and cellular homeostasis [45]. Integrin binds to the amino acid sequence of RACK1 between the fifth and seventh WD40 repeats, forming a protein complex. The binding sites for PKC are located between the third and fourth WD40 repeats, while PKCβ interacts with WD40-3 and WD40-5. In vertebrates, it has been demonstrated that RACK1 plays a crucial role in embryonic development and normal adult metabolism. This study investigated the expression characteristics of RACK1 during different developmental stages of zoea larvae from *E.carinicauda*. Our results indicate that *Ec*RACK1 is expressed in zoea larvae at stages 1 to 6, suggesting its close association with early development and metamorphosis in *E.carinicauda*.

Although the RACK1 gene has been extensively studied in vertebrates, its investigation in invertebrates remains limited. In recent years, the pivotal role of the RACK1 protein in safeguarding organisms against diverse environmental factors has emerged. In invertebrates, it was confirmed that the expression of RACK1 in the gonads of *M. arenaria* males exposed to TBT was significantly down-regulated (*p* < 0.05), suggesting that RACK1 may be a useful biomarker of exposure to TBT in the reproductive system of bivalve molluscs [16]. The expression of RACK1 in the hepatopancreas of *Pinctada martensii* exhibited a significant increase after 2 days of exposure to calcium at a concentration of 100 ng/L, and this upregulation continued to intensify over time [26]. In this study, we observed that the expression level of *Ec*RACK1 also increased in Aroclor 1254 treatment. During the experimental period, the expression of *Ec*RACK1 exhibited an initial increase followed by a subsequent decrease; however, it remained significantly elevated compared to the control group. RACK1 is a multifunctional protein involved in the regulation of antioxidant capacity and is closely associated with organismal growth and development [17,25]. Oxidative stress represents a common toxic mechanism induced by marine pollutants. In *E.carinicauda*, PCBs may induce oxidative stress, leading to the up-regulation of *Ec*RACK1 expression as a protective response against PCBs exposure. Furthermore, PCBs exert negative effects on the growth and development of marine organisms, while RACK1 plays a regulatory role in this process. Therefore, it is plausible that an increased expression of RACK1 enables white shrimp to counteract the detrimental impact of PCBs on their growth and development. However, the mechanism of RACK1 expression change under PCBs exposure needs further experiments to be proven.

## 5. Conclusions

Our results suggest that the length of the *Ec*RACK1 gene is 957 nucleotides, which encodes 318 amino acids. The molecular weight and theoretical pI of *Ec*RACK1 are approximately 35.6 kDa and 7.90, respectively. In addition, there are seven typical WD repeats in the secondary structure of *Ec*RACK1, which have more than a 96% sequence identity with the RACK1 protein of *Penaeus*. Meanwhile, *Ec*RACK1 was significantly increased in the zoea II and zoea IV stages, but had a significant tissue specificity in hepatopancreas, spermary, and muscle tissues of *E. carinicauda* in the adult stage. Under Aroclor 1254 exposure, *E*cRACK1 expression was significantly increased throughout the experimental period. These results suggest that *Ec*RACK1 may play an important role in the larval development and environmental defense of *E. carinicauda*. More extensive evidence related to the molecular function of RACK1 on crustaceans needs further study.

## Figures and Tables

**Figure 1 biology-13-00174-f001:**
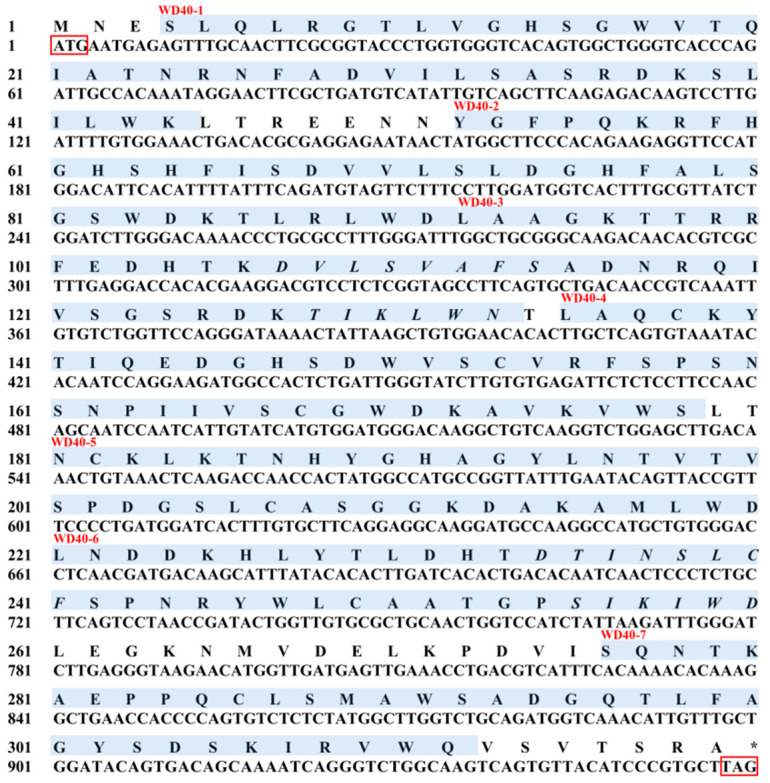
Nucleotides and deduced amino acid sequences of EcRACK1 from *Exopalaemon carinicauda*. The initiation codon (ATG) and stop codon (TAG) are boxed, respectively. Seven WD domains predicted by the SMART program are shaded in blue. “*” indicated the Amino acids for stop codon (TAG).

**Figure 2 biology-13-00174-f002:**
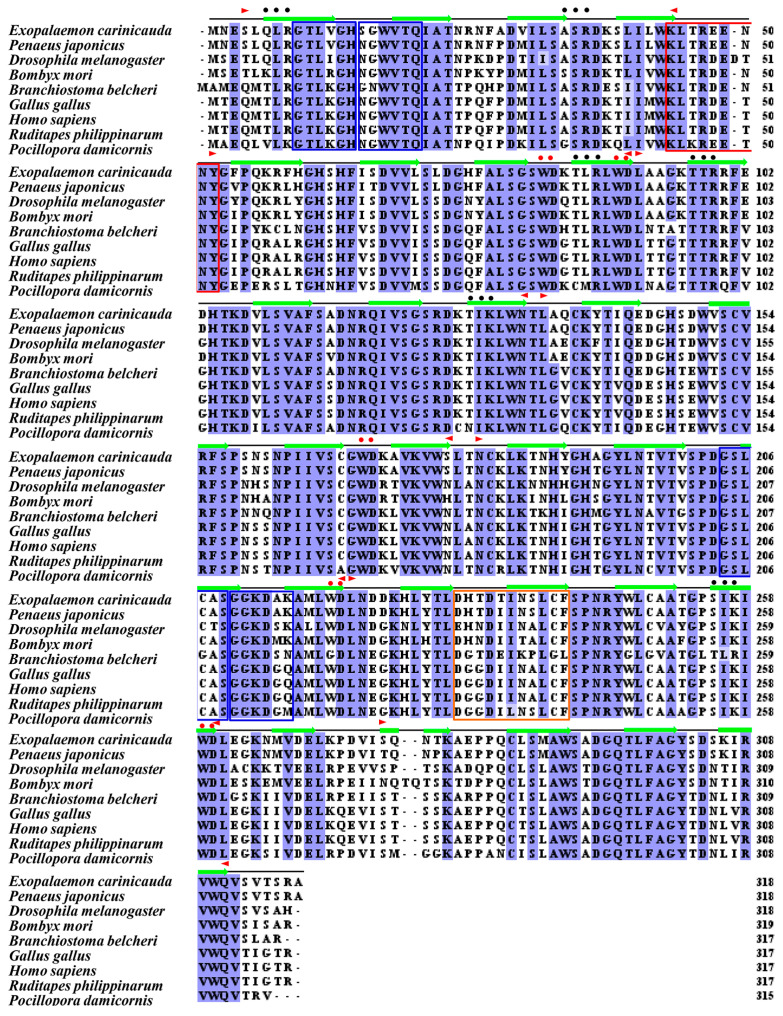
Multiple protein sequences alignment of RACK1 amino acid sequences and secondary structure predicted of WD40 domains for *Ec*RACK1 protein. RACK1 homologues used here were retrieved from NCBI database, and the species designations were listed on the left. Missing amino acids are shown as dashes. The conserved domains were predicted and are marked with purple shade in this figure. Black lines represent the amino acid sequence of *Ec*RACK1, in which green arrows are β-pleated sheets. Seven WD domains are indicated with the two opposite red triangles, respectively. Six PKC phosphorylation sites, one PKC activation site, four N-myristoylation sites, one tyrosine kinasephosphorylation site, and five WD motifs are marked with black dots, an orange box, blue boxes, a red box, and red dots, respectively.

**Figure 3 biology-13-00174-f003:**
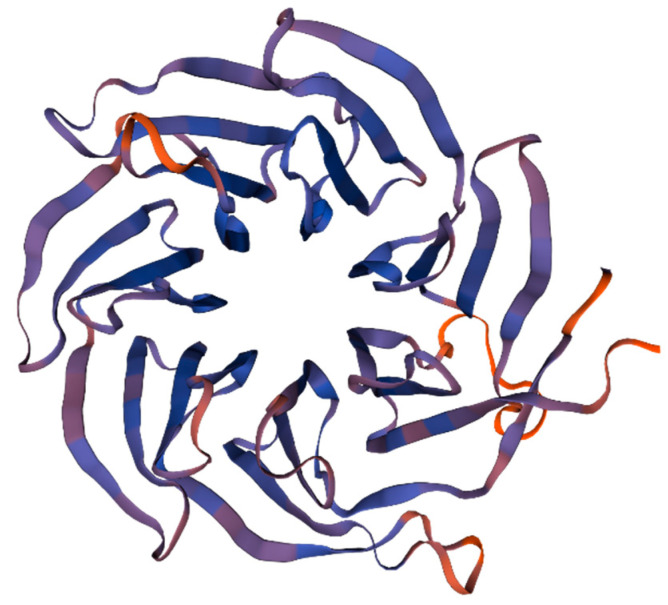
The model of WD40 domains of *Ec*RACK1 predicted according to the amino acid sequence.

**Figure 4 biology-13-00174-f004:**
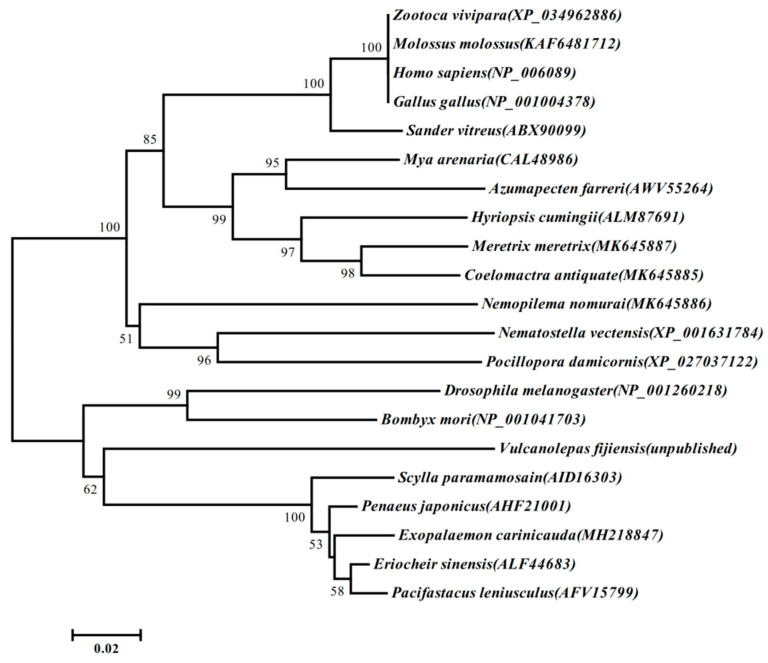
Phylogenetic tree based on homologous proteins of RACK1 found in other species by the MEGA 5.2 program. The phylogenetic tree was generated using the neighbour-joining method and the percentage concordance based on 10,000 bootstrap iterations is shown at the nodes. GenBank sequences in the tree include: *Exopalaemon carinicauda* RACK1 (MH218847); *Nemopilema nomurai* RACK1 (MK645886); *Coelomactra antiquate* RACK1 (MK645885); *Meretrix meretrix* RACK1 (MK645887); *Vulcanolepas fijiensis* RACK1 (PRJNA856037); *Eriocheir sinensis* RACK1 (ALF44683); *Scylla paramamosain* RACK1 (AID16303); *Pacifastacus leniusculus* RACK1 (AFV15799); *Penaeus japonicus* RACK1 (AHF21001); *Drosophila melanogaster* RACK1 (NP_001260218); *Bombyx mori* RACK1 (NP_001041703); *Gallus gallus* RACK1 (NP_001004378); *Homo sapiens* RACK1 (NP_006089); *Molossus molossus* RACK1 (KAF6481712); *Sander vitreus* RACK1 (ABX90099); *Zootoca vivipara* RACK1 (XP_034962886); *Hyriopsis cumingii* RACK1 (ALM87691); *Chlamys farreri* RACK1 (AWV55264); *Mya arenaria* RACK1 (CAL48986); *Nematostella vectensis* RACK1 (XP_001631784); and *Pocillopora damicornis* RACK1 (XP_027037122).

**Figure 5 biology-13-00174-f005:**
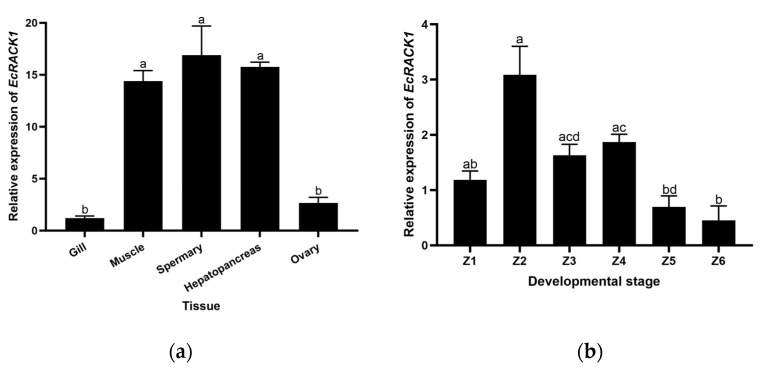
Characteristic of tissue distribution (**a**) and developmental expression (mean ± standard) of zoea larvae (**b**) of the *Ec*RACK1 gene of *Exopalaemon carinicauda* (N = 30). Note: different letters indicate statistically significant disparities between the two experimental groups.

**Figure 6 biology-13-00174-f006:**
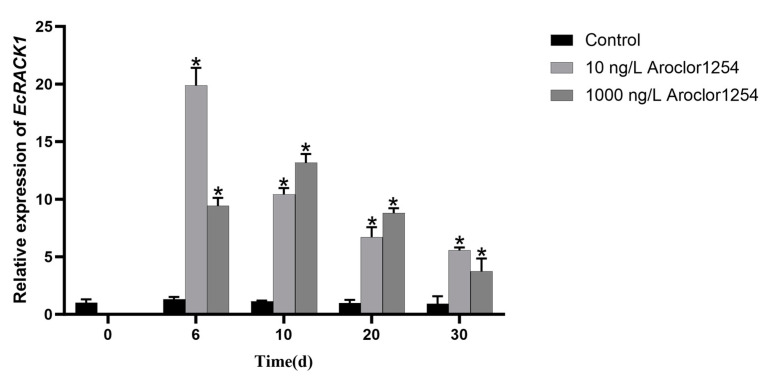
The expression levels of *Ec*RACK1 of zoea larvae of *Exopalaemon carinicauda* under Aroclor 1254 exposure. Asterisks (* *p* < 0.05) indicated that values were extremely significantly different from the control.

**Table 1 biology-13-00174-t001:** Primers used in the study.

Names	Sequences (5′–3′)	Efficiency
EcRACK1-F for ORF	ATGAATGAGAGTTTGCAACTTC	/
EcRACK1-R for ORF	CTAAGCACGGGATGTAACACT	
EcRACK1-qF	AGATTGCCACAAATAGGAACT	2.0982
EcRACK1-qR	GCACTGAAGGCTACCGAGA	
Ecβ-actin-qF	TGACGAAGACGCAACAGC	1.9116
Ecβ-actin-qR	TCATCGCCGACATAAGAG	

**Table 2 biology-13-00174-t002:** Location and amino acid sequences of seven WD in *Ec*RACK1 predicted by the SMART program.

Names	Sequences (C–N)
WD40-1	^4^SLQLRGTLVGHSGWVTQIATNRNFADVILSASRDKSLILWK^44^
WD40-2	^52^YGFPQKRFHGHSHFISDVVLSLDGHFALSGSWDKTLRLWD^91^
WD40-3	^92^LAAGKTTRRFEDHTKDVLSVAFSADNRQIVSGSRDKTIKLWN^133^
WD40-4	^135^LAQCKYTIQEDGHSDWVSCVRFSPSNSNPIIVSCGWDKAVKVWS^178^
WD40-5	^181^NCKLKTNHYGHAGYLNTVTVSPDGSLCASGGKDAKAMLWD^220^
WD40-6	^223^LNDDKHLYTLDHTDTINSLCFSPNRYWLCAATGPSIKIWD^260^
WD40-7	^276^SQNTKAEPPQCLSMAWSADGQTLFAGYSDSKIRVWQ^311^

## Data Availability

Sequences of *Ec*RACK1 gene from *Exopalaemon carinicauda* are available in GenBank with the accession number: MH218847.

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
