# Peer review of "Molecular Cloning, Characterization, and Expression of a Receptor for Activated Protein Kinase C1 (RACK1) Gene in Exopalaemon carinicauda Zoea Larvae under Aroclor 1254 Stress"

_biology, 2024, doi:10.3390/biology13030174_

Round 1

Reviewer 1 Report

Comments and Suggestions for Authors

This article investigates the structure and characteristics, tissue specificity and developmental stage gene expression pattern of RACK1 in Exopalaemon carinicauda larve. Meanwhile, the author also researched in the expression level of RACK1 in larval E. carinicauda exposed to Aroclor 1254. The integrated results showed that RACK1 could play an important role in larval development and environment defense of E. carinicauda. 

Overall, after careful reviewing, the authors performed a systematic experiment and interpreted their data in depth. The results and conclusions are convincing. Meanwhile, the topic of the paper is of interest for the scientific community and the paper is well-written and clear. However, the paper needs to be improved before acceptance for publication, as indicated below.

Line 94-169, the section should pay attention to the layout between numbers and units (no space between numbers and units.

Line 238-242, the sentence, “In the late stage of exposure, ……the late larval stage.”, was repeated. it is recommended to delete the repeat.

Line 334-335, in this sentence, “In crustaceans, …pearl oyster P. martensii RACK……”, the P. martensii is not belong to crustaceans.

Comments on the Quality of English Language

Minor editing of English language is required before publication. For example:

Line 47-48, the word in this sentence is incorrect, “With the regulating activity with protein kinase C (PKC), ……., has been the subject of growing concern and criticism.” The word “criticism” indicates negative effect. It is recommended to delete it.

Line 250-254, The sentence, “In organism, was a scaffold……”, lacked a subject.

Line 305-306, In this sentence, “In previous, ……. the gene in different biological conservatism…….”, the expression is not appropriate.

Author Response

Dear Reviewer 1:

 We thank you for providing constructive feedback. We have fully revised our manuscript and have addressed your comments.

This article investigates the structure and characteristics, tissue specificity and developmental stage gene expression pattern of RACK1 in Exopalaemon carinicauda larve. Meanwhile, the author also researched in the expression level of RACK1 in larval E. carinicauda exposed to Aroclor 1254. The integrated results showed that RACK1 could play an important role in larval development and environment defense of E. carinicauda.

Overall, after careful reviewing, the authors performed a systematic experiment and interpreted their data in depth. The results and conclusions are convincing. Meanwhile, the topic of the paper is of interest for the scientific community and the paper is well-written and clear. However, the paper needs to be improved before acceptance for publication, as indicated below.

Response: Thank you for your careful review of our manuscript and providing many

useful comments, we have carefully revised it according to your comment.

Major comments:

1. Line 94-169, “the section should pay attention to the layout between numbers and units (no space between numbers and units.”

Response 1: Thank you for your comment. The format of the numbers and units has been thoroughly looked over, and we've added spaces where there used to be no space between them.

2. Line 238-242, “the sentence, “In the late stage of exposure, ……the late larval stage.”, was repeated. it is recommended to delete the repeat.”

Response 2: Thank you for your suggestion. We have removed the duplicates.

3. Line 334-335, “in this sentence, “In crustaceans, …pearl oyster martensii RACK……”, the P. martensiiis not belong to crustaceans.”

Response 3: Thank you for your comment. We have revised this section.

The expression of RACK1 in the hepatopancreas of Pinctada martensii exhibited a significant increase after 2 days of exposure to calcium at a concentration of 100 ng/L, and this upregulation continued to intensify over time

Reviewer 2 Report

Comments and Suggestions for Authors

Review report on “Molecular cloning, characterization, and expression of a receptor for activated protein kinase C1 (RACK1) gene in Exopalaemon carinicauda mysis larvae under Aroclor 1254 stress”

A brief summary

The paper describes tissue-specific and developmental stage-specific gene expression of a scaffolding protein (RACK1) in a crustacean species and the effects of exposure to an environmental toxin on expression of this gene. The gene is sequenced and quantitative PCR is used to examine the effects on larvae, although the results are not well described. Further experiments on effects of the increased gene expression on physiology, growth, development would provide more useful information on whether Acroclor is harmful to this species or the effects on other genes as well as RACK1. It is difficult to understand the rationale behind the study simply by examining the expression of one gene which assists so many others in their function.

General comments

First of all, language let this manuscript down, since there are many areas that were not described clearly, making the experiments and results difficult to understand. Rewording many areas of the manuscript must be carried out. The concept of exposure to Aroclor is interesting but the authors do not describe why it is necessary to study in this species, or  why it is important to understand the effects on the specific protein investigated. The summary, abstract and conclusion are identical and so they must be rewritten. The authors describe a reduction in RACK1 expression at 20 days but this is not apparent from their data: the written description of their results differs from the qPCR graph shown. In the conclusion section (and abstract and summary) the authors need to describe why they think that “EcRACK1 may play an important role in larval development and environmental defense….” as I feel further experiments would be required before such a statement can be made. The authors didn’t measure any parameters concerning larval development or environmental defence, only gene expression of one protein that has many functions.

Specific comments

Line 16 and 32: Do you mean <0.05?

Line 17 and 33: What is a spermary?

Line 20 and 39: What do you mean “more favorable”?

Line 45: Reword this sentence -  “With the regulating activity with protein kinase C, scaffolding protein receptor for activated protein kinase C1….. etc”. Too many “withs” and unclear.

Line 47: why is RACK1 a concern? Why is it the subject of criticism? Please reword or explain more clearly what you mean.

Line 55: Remove “As so for,” change to “have” from “had”

Line 61: “Conservative”?

Throughout: check for plurals and appropriate use of has or had

Line 74: which suggested that instead of deduced

Line 74: What does “silently-expressed” mean? Do you mean that the gene was silenced?

Line 78 change Bap to BaP

Line 82: coastal what?

Line 86: why did you use larvae instead of adults? Surely it would be easier to use adults for tissue distribution etc?

Line 88: Please include in the introduction why you were interested in Aroclor in particular. Is exposure to Aroclor a major problem? Is it in the environment and at what concentrations has it been found? Have there been other studies investigating the effects of Aroclor on crustaceans or other species? More information is required for the purpose of the study.

Line 93 – 97: please check the grammar.

Line 99: Did you use adults or larvae to obtain tissues? The introduction says larvae, here you only mention adults being collected and maintained.

Line 102: how did you homogenise the tissues?

Figure 2: The species names are unclear. What do the green arrows and red boxes show? A key is required or more information in the figure legend please.

Table 3 could be moved to the Supplementary Information.

Line 217: hepatopancreas

Line 217: Hepatopancreas didn’t show the highest levels according to Figure 5a: it was the same as muscle, spermary (what is spermary? Is this testis?).

Figure 5: Please put the sample size (N) in the figure legend – how many were in each group? Are you showing mean and standard error or standard deviation? What do z1 to z6 mean? What are their ages/sizes?

Figure 6: What was your control? Did it contain DMSO? If so, what concentration of DMSO was in the control?

Line 228: Where is the significant reduction at 20 days shown? I can only see high expression compared with controls for each time point. Is there a significant difference between the two concentrations used?

Line 233: What do you mean by “….. because all mysis larvae entered the late larval stage”. What does this show? Also, do you have data – numbers of animals in different developmental stages to present? Were there any differences in the appearance of the larvae exposed to Aroclor – do you have any photographs of the larvae if there were differences in appearance?

Discussion: Please correct the language inaccuracies throughout.

Line 303: I think that NHE5 controls ionic balance, not metabolic balance, please check if this is correct.

Line 332: which results obtained on mRNA?

Line 335-336: Your results didn’t show this at all – Figure 6 didn’t show your Aroclor exposed animals returning to normal expression levels of RACK1 at all. Please clarify.

Line 339-340: I don’t see any evidence of an acceleration in the metamorphosis of the late larvae – please provide images or data to support this statement or remove it.

Line 340-341: This doesn’t make sense.

Conclusions: These are the same as the abstract and summary and therefore should be changed.

In the discussion, please describe why you think that the RACK1 gene expression was increased following exposure to Aroclor.

Comments on the Quality of English Language

Review report on “Molecular cloning, characterization, and expression of a receptor for activated protein kinase C1 (RACK1) gene in Exopalaemon carinicauda mysis larvae under Aroclor 1254 stress”

A brief summary

The paper describes tissue-specific and developmental stage-specific gene expression of a scaffolding protein (RACK1) in a crustacean species and the effects of exposure to an environmental toxin on expression of this gene. The gene is sequenced and quantitative PCR is used to examine the effects on larvae, although the results are not well described. Further experiments on effects of the increased gene expression on physiology, growth, development would provide more useful information on whether Acroclor is harmful to this species or the effects on other genes as well as RACK1. It is difficult to understand the rationale behind the study simply by examining the expression of one gene which assists so many others in their function.

General comments

First of all, language let this manuscript down, since there are many areas that were not described clearly, making the experiments and results difficult to understand. Rewording many areas of the manuscript must be carried out. The concept of exposure to Aroclor is interesting but the authors do not describe why it is necessary to study in this species, or  why it is important to understand the effects on the specific protein investigated. The summary, abstract and conclusion are identical and so they must be rewritten. The authors describe a reduction in RACK1 expression at 20 days but this is not apparent from their data: the written description of their results differs from the qPCR graph shown. In the conclusion section (and abstract and summary) the authors need to describe why they think that “EcRACK1 may play an important role in larval development and environmental defense….” as I feel further experiments would be required before such a statement can be made. The authors didn’t measure any parameters concerning larval development or environmental defence, only gene expression of one protein that has many functions.

Specific comments

Line 16 and 32: Do you mean <0.05?

Line 17 and 33: What is a spermary?

Line 20 and 39: What do you mean “more favorable”?

Line 45: Reword this sentence -  “With the regulating activity with protein kinase C, scaffolding protein receptor for activated protein kinase C1….. etc”. Too many “withs” and unclear.

Line 47: why is RACK1 a concern? Why is it the subject of criticism? Please reword or explain more clearly what you mean.

Line 55: Remove “As so for,” change to “have” from “had”

Line 61: “Conservative”?

Throughout: check for plurals and appropriate use of has or had

Line 74: which suggested that instead of deduced

Line 74: What does “silently-expressed” mean? Do you mean that the gene was silenced?

Line 78 change Bap to BaP

Line 82: coastal what?

Line 86: why did you use larvae instead of adults? Surely it would be easier to use adults for tissue distribution etc?

Line 88: Please include in the introduction why you were interested in Aroclor in particular. Is exposure to Aroclor a major problem? Is it in the environment and at what concentrations has it been found? Have there been other studies investigating the effects of Aroclor on crustaceans or other species? More information is required for the purpose of the study.

Line 93 – 97: please check the grammar.

Line 99: Did you use adults or larvae to obtain tissues? The introduction says larvae, here you only mention adults being collected and maintained.

Line 102: how did you homogenise the tissues?

Figure 2: The species names are unclear. What do the green arrows and red boxes show? A key is required or more information in the figure legend please.

Table 3 could be moved to the Supplementary Information.

Line 217: hepatopancreas

Line 217: Hepatopancreas didn’t show the highest levels according to Figure 5a: it was the same as muscle, spermary (what is spermary? Is this testis?).

Figure 5: Please put the sample size (N) in the figure legend – how many were in each group? Are you showing mean and standard error or standard deviation? What do z1 to z6 mean? What are their ages/sizes?

Figure 6: What was your control? Did it contain DMSO? If so, what concentration of DMSO was in the control?

Line 228: Where is the significant reduction at 20 days shown? I can only see high expression compared with controls for each time point. Is there a significant difference between the two concentrations used?

Line 233: What do you mean by “….. because all mysis larvae entered the late larval stage”. What does this show? Also, do you have data – numbers of animals in different developmental stages to present? Were there any differences in the appearance of the larvae exposed to Aroclor – do you have any photographs of the larvae if there were differences in appearance?

Discussion: Please correct the language inaccuracies throughout.

Line 303: I think that NHE5 controls ionic balance, not metabolic balance, please check if this is correct.

Line 332: which results obtained on mRNA?

Line 335-336: Your results didn’t show this at all – Figure 6 didn’t show your Aroclor exposed animals returning to normal expression levels of RACK1 at all. Please clarify.

Line 339-340: I don’t see any evidence of an acceleration in the metamorphosis of the late larvae – please provide images or data to support this statement or remove it.

Line 340-341: This doesn’t make sense.

Conclusions: These are the same as the abstract and summary and therefore should be changed.

In the discussion, please describe why you think that the RACK1 gene expression was increased following exposure to Aroclor.

Reviewer 3 Report

Comments and Suggestions for Authors

The parts that need correction are specified in the attached file. Please check and reflect it in your manuscript.

Author Response

Dear reviewer 3:

We thank you for providing constructive feedback. We have fully revised our manuscript and have addressed your comments.

  1. Please describe the analysis method in more detail in the abstract.

Response 1: Thank you for your comment. We have added more details of the analytical methods in the abstract.

 In a previous study, we identified the RACK1 sequence of white shrimp from transcriptome data. In this study, we employed specialized bioinformatics software to conduct an in-depth analysis of EcRACK1 and compare its amino acid sequence homology with other crustaceans. Furthermore, we investigated the expression patterns of RACK1 at different developmental stages and tissues, as well as at various time points after exposure to Aroclor 1245, aiming to elucidate its function and po-tential response towards Aroclor 1245 exposure.

  1. What is the basis for criticism?

Response 2: Thank you for your comment.  We misused the word and have revised the sentence.

The widespread presence of scaffolding protein receptor for activated protein ki-nase C1 (RACK1) in both prokaryotes and eukaryotes, along with its regulatory activi-ty on protein kinase C (PKC), has garnered increasing attention and scrutiny

  1. The introduction section contains the purpose and content of the future research. Therefore, the entire sentence itself is not written in the past tense. Please clarify the purpose of this part and describe the content of your research.

Response 3:  Thank you for your comment. We have revised it in the manuscript.

Exopalaemon carinicauda is widely distributed in coastal areas and belongs to the benthic shrimp species. Due to its rapid growth, strong reproductive capacity, euryhaline nature, and ease of feeding, it has become a prominent shrimp species bred in recent years. This study presents the gene characteristics, tissue distribution, and developmental expression pattern of EcRACK1 in E. carinicauda mysis larvae. Furthermore, to explore its potential function in the metabolic pathways of exogenous substances in invertebrates, we measured the mRNA levels of EcRACK1 after exposure to Aroclor 1254. Our research enhances understanding of the characterization and multiple functions of RACK1 in invertebrates.

  1. Please change the unit to psu.

Response 4: Thank you for your suggestion. We have changed the salinity unit to psu.

  1. Delete the dot in the table title. Please reflect this throughout the paper.

Response 5: Thank you for your suggestion. We have deleted the dot in the table title.

  1. Please describe the object of analysis in more detail.

Response 6:  Thank you for your comment. We have added more details of the analytical object.

The data analysis was performed using SPSS Statistics 25 (SPSS Inc.). RACK1 ex-pression was quantified using the 2-△△Ct method. All results are presented as means ± SE. One-way analysis of variance with the LSD method was employed for intergroup comparisons. Statistical significance was defined as P < 0.05.

  1. The results of the ANOVA test for three groups (control, 10ng/L, 1000ng/L) were displayed for each time. If you look at the manuscript, statistical analysis was also conducted between Time (d) groups. Please also indicate this part in the picture.

Response 7:  Thank you for your comment. It may be that we were not articulate, but we did not make comparisons between time groups in this section.

  1. The general functions of RACK1 should be mentioned in the introduction, and please consider additional studies that analyzed RACK1 in aquatic organisms.

Response 8:  We have made corresponding modifications, and the proposed research content will be explored in our subsequent relevant studies.

Reviewer 4 Report

Comments and Suggestions for Authors

The authors presented their research to test the hypothesis that RACK1 plays a crucial role in the larval development and environmental defense of Exopalaemon carinicauda. RACK1 is recognized as an adaptor protein involved in various intracellular signal transduction pathways, contributing to diverse biological processes such as cell proliferation, migration, apoptosis, development, cellular adaptation, and immune response. Initially studied in multiple aquatic invertebrates, RACK1 has demonstrated its multifunctionality.

To examine their hypothesis thoroughly, the authors conducted two aspects of research. Firstly, the authors analyzed the structural characteristics, tissue specificity, and gene expression patterns at different developmental stages of RACK1 in E.carinicauda larvae. Secondly, the authors investigated the expression level of RACK1 in E.carinicauda larvae after exposure to Aroclor 1254. This study provides initial insights into exploring the role of RACK1 in larval development and environmental defense mechanisms of E.carinicauda.

After a thorough review, the authors conducted a systematic experiment and provided in-depth interpretation of their data, resulting in convincing results and conclusions. Additionally, the paper's topic is of interest to the scientific community and is well-written and clear. However, some improvements are needed before this paper can be accepted for publication.

Section 2.1, how to handle the samples during E.carinicauda larval development period?

Line 94-169, some numbers and units are incorrectly formatted without spaces.

Line 228-230, Figure 5 don’t provide no labeling information. What do the alphabet in the picture mean?

Line 238-242, the section contains duplicates.

Line 305-306, mentions previous studies, but no references are given.

Author Response

Dear reviewer 4:

We thank you for providing constructive feedback. We have fully revised our manuscript and have addressed your comments.

The authors presented their research to test the hypothesis that RACK1 plays a crucial role in the larval development and environmental defense of Exopalaemon carinicauda. RACK1 is recognized as an adaptor protein involved in various intracellular signal transduction pathways, contributing to diverse biological processes such as cell proliferation, migration, apoptosis, development, cellular adaptation, and immune response. Initially studied in multiple aquatic invertebrates, RACK1 has demonstrated its multifunctionality.

To examine their hypothesis thoroughly, the authors conducted two aspects of research. Firstly, the authors analyzed the structural characteristics, tissue specificity, and gene expression patterns at different developmental stages of RACK1 in E.carinicauda larvae. Secondly, the authors investigated the expression level of RACK1 in E.carinicauda larvae after exposure to Aroclor 1254. This study provides initial insights into exploring the role of RACK1 in larval development and environmental defense mechanisms of E.carinicauda.

After a thorough review, the authors conducted a systematic experiment and provided in-depth interpretation of their data, resulting in convincing results and conclusions. Additionally, the paper's topic is of interest to the scientific community and is well-written and clear. However, some improvements are needed before this paper can be accepted for publication.

  1. Section 2.1, how to handle the samples during E.carinicaudalarval development period?

Response 1: All the experimental samples underwent a unified microscopic examination

  1. Line 94-169, some numbers and units are incorrectly formatted without spaces.

Response 2: Thank you for your commnent. The format of the numbers and units has been thoroughly looked over, and we've added spaces where there used to be no space between them.

  1. Line 228-230, Figure 5 don’t provide no labeling information. What do the alphabet in the picture mean?

Response 3: Thank you for your comment. The presence of distinct letters indicates statistically significant disparities between the two experimental groups.

  1. Line 238-242, the section contains duplicates.

Response 4: Thank you for your comment. We have removed duplicates from the manuscript.

  1. Line 305-306, mentions previous studies, but no references are given.

Response 5: Thank you for your comment. We have added literature in this section, as the following list.

Buoso E.; Masi M.; Galbiati V.; Maddalon A.; Iulini M.; Kenda M.; Dolenc M.S.; Marinovich M.; Corsini E. Effect of estrogen-active compounds on the expression of RACK1 and immunological implications. Archives of Toxicology 2020, 94, 2081-2095.

Nagashio R.; Sato Y.; Matsumoto T.; Kageyama T.; Satoh Y.; Shinichiro R.; Masuda N.; Goshima N.; Jiang S.; Okayasu I. Expression of RACK1 is a novel biomarker in pulmonary adenocarcinomas. Lung Cancer 2010, 69, 54-59.

Song Y Y.; Zhang X Z.; Wang B N.; Cheng Y K.; Guo X.; Zhang X.; Shao R L.; Liu R D.; Wang Z Q.; Cui J. A novel Trichinella spiralis serine proteinase disrupted gut epithelial barrier and mediated larval invasion through binding to RACK1 and activating MAPK/ERK1/2 pathway. PLoS neglected tropical diseases, 2024, 18(1), e0011872-e0011872.

Round 2

Reviewer 2 Report

Comments and Suggestions for Authors

Second Review report on “Molecular cloning, characterization, and expression of a receptor for activated protein kinase C1 (RACK1) gene in Exopalaemon carinicauda mysis larvae under Aroclor 1254 stress”

Specific comments

Line 75: Please change from “silently-expressed”. A gene cannot be silently-expressed, they are opposites. Please change the sentence to say that the gene was silenced. Thankyou.

Line 89: Your new section has the reference 27 (Cappello T. et al 2016) is for mercury-induced effects, perhaps consider finding another reference.

Line 94: Please change from Marine organisms to marine organisms. Same for Line 108.

Line 256-7: You did not show highest levels of gene expression in the hepatopancreas, it was the same as muscle, spermary (all have the same a above them so not significantly different). Please remove the sentence.

Figure 5: Please put the sample size (N) in the figure title N = 30. It is very helpful to read it in the Figure legend without having to look back through the methods. Please also state in the Figure titles that Mean and Standard deviation are shown, and that different letters indicate significantly different results. Thankyou.

Line 266: Spelling error RACK1. Please also remove the word “gradual”. The expression went straight from 1 to 20 at day 6 and then decreased over the following days. You could write that instead.

Line 271: Thankyou for providing your photographs of larvae. They are very interesting.

Thankyou for modifying the discussion to describe your reasons why RACK1 expression is increased following Aroclor exposure.

Conclusion: Please change the word “favorable” to “extensive” perhaps or just remove the word “favorable”. Thankyou. Also change “crustacean” to “crustaceans”. Thanks.

Comments on the Quality of English Language

Second Review report on “Molecular cloning, characterization, and expression of a receptor for activated protein kinase C1 (RACK1) gene in Exopalaemon carinicauda mysis larvae under Aroclor 1254 stress”

Specific comments

Line 75: Please change from “silently-expressed”. A gene cannot be silently-expressed, they are opposites. Please change the sentence to say that the gene was silenced. Thankyou.

Line 89: Your new section has the reference 27 (Cappello T. et al 2016) is for mercury-induced effects, perhaps consider finding another reference.

Line 94: Please change from Marine organisms to marine organisms. Same for Line 108.

Line 256-7: You did not show highest levels of gene expression in the hepatopancreas, it was the same as muscle, spermary (all have the same a above them so not significantly different). Please remove the sentence.

Figure 5: Please put the sample size (N) in the figure title N = 30. It is very helpful to read it in the Figure legend without having to look back through the methods. Please also state in the Figure titles that Mean and Standard deviation are shown, and that different letters indicate significantly different results. Thankyou.

Line 266: Spelling error RACK1. Please also remove the word “gradual”. The expression went straight from 1 to 20 at day 6 and then decreased over the following days. You could write that instead.

Line 271: Thankyou for providing your photographs of larvae. They are very interesting.

Thankyou for modifying the discussion to describe your reasons why RACK1 expression is increased following Aroclor exposure.

Conclusion: Please change the word “favorable” to “extensive” perhaps or just remove the word “favorable”. Thankyou. Also change “crustacean” to “crustaceans”. Thanks.

Author Response

Dear reviewer 2:

We appreciate your constructive feedback again. We have completely revised our manuscript and addressed your comments.

Specific comments

  1. Line 75: Please change from “silently-expressed”. A gene cannot be silently-expressed, they are opposites. Please change the sentence to say that the gene was silenced. Thankyou.

Response 1: Thank you for your suggestion. We have changed from “silently-expressed”.

In M. coruscus, RACK1 expression was silenced, which resulted in increased activities of superoxide dismutase (SOD) and catalase (CAT) in haemocytes, suggesting a crucial role for RACK1 in the oxidative stress response of M. coruscus.

  1. Line 89: Your new section has the reference 27 (Cappello T. et al 2016) is for mercury-induced effects, perhaps consider finding another reference.

Response 2: Thank you for your comment. We have replaced this reference in the manuscript.

[27] Sardenne F.; Loc'h L F.; Bodin N. Persistent organic pollutants and trace metals in selected marine organisms from the Akanda National Park, Gabon (Central Africa). Marine Pollution Bulletin, 2024, 199, 199116009.

  1. Line 94: Please change from Marine organisms to marine organisms. Same for Line 108.

Response 3:  Thank you for your suggestion.  We have changed from “Marine organisms” to “marine organisms”.

  1. Line 256-7: You did not show highest levels of gene expression in the hepatopancreas, it was the same as muscle, spermary (all have the same a above them so not significantly different). Please remove the sentence.

Response 4:  Thank you for your suggestion. We have removed it in the manuscript.

  1. Figure 5: Please put the sample size (N) in the figure title N = 30. It is very helpful to read it in the Figure legend without having to look back through the methods. Please also state in the Figure titles that Mean and Standard deviation are shown, and that different letters indicate significantly different results. Thankyou.

Response 5: Thank you for your comment. We have added it in the manuscript.

  1. Line 266: Spelling error RACK1. Please also remove the word “gradual”. The expression went straight from 1 to 20 and at day 6 then decreased over the following days. You could write that instead.

Response 6:  Thank you for your suggestion.  We have revised it in the manuscript.

Compared to the control group, the expression went straight from 1 to 6 day and then decreased over the following days (P<0.05).

  1. Line 271: Thank you for providing your photographs of larvae. They are very interesting.

Response 7: Thank you for your comment. I am very glad that our modifications have attracted your interest.

  1. Thank you for modifying the discussion to describe your reasons why RACK1 expression is increased following Aroclor exposure.

Response 8: Thank you for your comment. I still appreciate your suggestion.

  1. Conclusion: Please change the word “favorable” to “extensive” perhaps or just remove the word “favorable”. Thankyou. Also change “crustacean” to “crustaceans”. Thanks.

Response 9: Thank you for your comment. We have revised it in the manuscript.

Reviewer 3 Report

Comments and Suggestions for Authors

The pointed out parts have been appropriately corrected throughout the manuscript.

Author Response

Thank you for your comment. I still appreciate your suggestion.